# A Longitudinal Analysis of a Motor Skill Parameter in Junior Triathletes from a Wearable Sensor

**DOI:** 10.3390/s26010096

**Published:** 2025-12-23

**Authors:** Stuart M. Chesher, Dale W. Chapman, Bernard Liew, Simon M. Rosalie, Hugh Riddell, Paula C. Charlton, Kevin J. Netto

**Affiliations:** 1Curtin School of Allied Health, Curtin University, Bentley, Perth, WA 6102, Australia; chesher_stuart@hotmail.com (S.M.C.);; 2School of Sport, Rehabilitation, and Exercise Sciences, University of Essex, Colchester CO4 3SQ, Essex, UK; 3SR Performance, Gairloch Drive, Frankston, VIC 3199, Australia; 4Australian Institute of Sport, Leverrier Street, Canberra, ACT 2617, Australia

**Keywords:** inertial measurement units, skill development, adolescent athletes, high performance pathway, coaching

## Abstract

**Purpose:** Optimal movement cadence is critical to success in elite triathlons. Therefore, the objective of this research was to investigate group and individual longitudinal changes in movement cadence amongst a group of junior triathletes. **Method:** Junior triathletes (season 1: n = 4, season 2: n = 11) who were members of the state’s talent development pathway wore a single trunk-mounted inertial measurement unit during triathlon races across two triathlon seasons (October 2021 to April 2023). Sensor data were analysed using both linear and non-linear modelling to identify changes in movement cadence across the three disciplines of the triathlon. This allowed for the differences between the two modelling techniques to be contrasted. A custom automatic peak detection algorithm was used to process and analyse the movement cadence data for each triathlete in each discipline. **Results:** Non-linear modelling performed significantly better than linear modelling in swimming; however, there were no significant differences in model performance between cycling and running. At a group level, non-linear modelling predicted increases in swimming and running cadence across the seasons. However, negligible changes were observed in cycling cadence across the same period. **Conclusions:** Meaningful changes in movement cadence can be detected with a single inertial measurement unit and confidently predicted in swimming and running over a competitive season when using non-linear modelling techniques. This approach reflects the non-linear nature of human motor skill development and paves the way for similar applications in other sports.

## 1. Introduction

The skilful control of coordinated movement is the subject of contemporary and historical scientific investigation. This is particularly relevant in sport, as athletes who demonstrate greater mastery of important motor skills often perform better than less skilled counterparts [1]. One method of investigating important skills in a sport is to explore the beliefs, attitudes and experiences of stakeholders, including expert coaches and elite athletes [2,3]. Once important skills are established, they can be quantitatively explored to determine how performance changes over time [2,3]. The quantification of various movement skills has become more ubiquitous in recent times due to the proliferation of micro-sensors and the expansive use of inertial measurement units (IMUs) in sports [4,5,6,7]. More specifically, the longitudinal measurement of athletic performance using wearable IMUs is providing a more comprehensive understanding of changes in sports performance over time [8,9].

To predict how motor skill performance changes over time based on observed data, a common method of modelling changes in motor skill performance is to use linear mixed-effects models or other similar linear analysis approaches [10,11,12]. However, investigations that measure changes in motor skill performance observe that such changes over time do not form a linear trend [13,14]. Thus, taking a similar methodological approach but using a non-linear statistical modelling analysis, such as generalised additive modelling [15], non-linear mixed-effects models or growth curve modelling [16], to correctly satisfy statistical assumptions would provide a more theoretically correct method to understand how motor skill performance changes over time.

In triathlons, the methodological approach of interviewing experts to identify important skills in short-distance triathlons (Olympic distance and sprint distance) was adopted by Chesher et al. [3], where stakeholders were interviewed about the motor skills that are important for success at the elite level. The investigation identified the kinematics of locomotion, posture, stabilisation, navigation and breathing, and the ability to adapt these skills to situational demands, which were all important elements of the continuous motor skills required for elite performance [3]. When an athlete varies the performance of a motor skill to suit the situational demands that it is performed under, one strategy used is known as parameterisation [17]. Parameterisation can be identified when the features of skill performance remain relatively the same, but the generalised motor program (GMP) is altered by changing the total speed or force with which it is performed. For continuous motor skills like swimming, cycling and running, parametrising the speed of movement alters the movement cadence (propulsive movements per unit of time). Performing longitudinal analyses of movement cadence allows for the detection of accelerations, decelerations and stabilisations in motor skill performance. This can provide a basis for applying appropriate training interventions in response to these trends [18].

Objectively assessing motor skills in triathlons requires a method of measuring important performance metrics during training and races. Wearable sensors have a variety of applications in measuring aspects of the physical world [19], including as a valid tool to measure a range of important motor skills in triathlons [20]. These sensors function by collecting movement information from the body segments that they are attached to, allowing the estimation of gross human movement. Thus, wearable sensor systems can be highly complex, containing multiple integrated sensors on many body parts, or simple, when utilising a single IMU. Additionally, wearable IMU data can be processed automatically by data science techniques to provide timely motor skill analyses of multiple triathletes [21]. This was shown when Chesher et al. [21] created a custom peak counting algorithm to automate the detection of swimming, cycling and running cadences from a single trunk-worn IMU in a triathlon. This algorithm combined accelerometer and gyroscope signals that were filtered using a Butterworth filter and counted using ‘SciPy’ (Python, version 3.12.0b3), with a 0.25–0.5-s minimum detection interval. In other investigations of automatic cadence analysis, swimming cadence was also analysed using time-series signal processing [22], whereas cycling cadence was analysed using a supervised machine learning approach [23].

While the richness of information provided by multi-sensor wearable systems can be desirable, they require advanced computer processing to integrate the information and are impractical to wear while racing. Therefore, this investigation has two aims: (1) to use a single IMU to measure movement cadence in the individual subdisciplines of a triathlon (swimming, cycling and running) and (2) to model these data using linear and non-linear general modelling to identify the expected change in movement cadence in each of the subdisciplines in a cohort of junior triathletes across two seasons of triathlon racing. We hypothesise that the change in movement cadence for each triathlon discipline will take a non-linear pattern and, thus, non-linear modelling will provide a more accurate model fit that offers a better prediction of performance changes over time.

## 2. Materials and Methods

### 2.1. Study Design

A longitudinal, prospective, observational study design was used to measure the swimming, cycling and running movement cadences of 12 triathletes (season 1: n = 4, season 2: n = 11) (competition level: nine tier 3, highly trained/national, and three tier 2, trained/developmental [24]) during races over two triathlon seasons spanning from October 2021 to April in 2023. A sample of 12 participants represented 32% of the region’s junior talent development pathway. Participant recruitment occurred in two cohorts. Before the 2021–2022 triathlon season, four junior triathletes (mean age = 16.32 ± 0.57 years; sex = three male and one female; average time in sport = 4.00 ± 0.79 years) were recruited from a single triathlon club. To participate, triathletes had to be at least 12 years old, free from injury and intending to participate in a full competitive triathlon season. Following this, a second round of recruitment occurred before the start of the 2022–2023 triathlon season, recruiting an additional eight triathletes from a different triathlon club (mean age = 15.88 ± 0.98 years; sex = three female and five male; average time in sport = 2.99 ± 1.69 years), while continuing to collect data from the original four triathletes. During the second recruitment period, one original participant withdrew from the sport; however, their data remained in the analysis (Figure 1). Ethical approval was obtained from the university’s human research ethics committee (HRE2021-0071).

### 2.2. Procedures

A wearable IMU (Catapult Optimeye S5, Catapult Sports Pty Ltd., Melbourne, Australia) containing a global positioning system (GPS) (10 Hz), tri-axial accelerometer and gyroscope sampling at 100 Hz was attached to each participant in a custom-made pouch pinned underneath the triathlon race suit between the shoulder blades. IMU calibration was performed by the manufacturer and was therefore not repeated manually prior to recording. The IMU has been previously validated to measure swimming, cycling and running cadences in triathlons via an automatic cadence detection program (swimming strokes = 98.7 ± 0.5% accuracy; cycling pedal strokes = 97.8 ± 0.9% accuracy; running strides = 99.4 ± 0.6% accuracy) [20,21,25,26]. Participants wore the IMU during triathlon races throughout two triathlon seasons. On one instance, there were multiple races on the same course in the same day. As the intention was to measure longitudinal changes in movement cadence due to improvements in motor skill performance, rather than acute changes (i.e., unmeasured fatigue), the performance in these races was averaged to a single result for each participant.

During the investigation, data were collected during 19 triathlons. Performance impacts of each discipline have been observed on the subsequent discipline [27,28]; thus, data were only included if the preceding swim or cycle was also included. On one occasion, a last-minute decision was made by the race organiser to change a triathlon to a duathlon (run–cycle–run) due to unsafe water conditions; thus, only the final run could be included from this race. Data were excluded from races for four reasons: when participants raced while unwell; when they were injured during the race; improper attachment of the IMU; and the GPS failing to connect to satellites. For each race, the time taken for the participant to complete the race was recorded, along with the GPS distance, to calculate the average race velocity. Swimming distances were recorded based on the race organiser’s directions, as the GPS was unavailable when the IMU was submerged. On eight occasions, the GPS malfunctioned and was unable to accurately locate the position of the athlete, causing large and rapid changes in the recorded global position that were outside the plotted racecourse. When this occurred, a close approximation of the distance was calculated by averaging the race distance obtained from the functioning GPS sensors attached to other research participants in the same race. This resulted in 83 data points for swimming (season one: 22 data points, season two: 61 data points) and cycling (season one: 27 data points, season two: 56 data points) and 86 data points for running (season one: 31 data points, season two: 55 data points).

### 2.3. Statistical Analysis

Data from each IMU were processed using manufacturer-supplied software (Catapult Sprint 5.1.7, Catapult Sports Pty Ltd., Melbourne, Australia) and exported to a spreadsheet (Excel 2019, Microsoft Corporation, Redmond, WA, USA) for further processing and collation. A custom peak counting algorithm was then used to analyse the IMU data [21]. This algorithm was designed to analyse and count peaks and troughs in the medio-lateral axis of an accelerometer to detect swimming strokes. It also combined peaks and troughs on the yaw axis of a gyroscope and checked these against the antero-posterior axis of a synchronised accelerometer to detect pedal strokes in cycling. Finally, the algorithm detects peaks in the vertical axis of an accelerometer in running [20]. The algorithm also used a minimum detection interval of between 0.25 and 0.5 s, depending on the discipline, to improve the accuracy and prevent counting artifact peaks and troughs in the signal as strokes, pedals and strides. A graphical representation of this process can found in Chesher et al. [20]. The data were then imported into a statistical analysis program (R, 4.3.3, Posit, Boston, MA, USA) to analyse changes in performance over time. Group and individual average movement cadences (swimming strokes, pedal strokes or running strides per minute) for each race were then plotted against longitudinal time to visually assess the direction of the change in performance.

To answer the research questions, the distribution types of swimming, cycling and running cadences were checked via Kolmogorov–Smirnov tests of normality. As the assumption of normality was satisfied, the average movement cadence from each discipline was analysed separately using two Gaussian generalised additive models (GAMs) (linear and non-linear) with restricted maximum likelihood (REML) estimation [29].

For both analyses, the week number (*W*) was modelled as a predictor of the average cadence (*C*) and the season was included as a categorical fixed effect. Participants were modelled using random intercepts, and individual change over time was captured using random slopes of the week number. Non-linear trends in week number were modelled using penalised smooth functions. Random intercepts were included for each participant, with random slopes for intra-individual change over time. Let i index athlete (participant) and t index observation (week). The model equations for each analysis method were as follows:

Linear model: *C_it_* = *β*_0_ + *β*_1_ *W_it_* + *β*_2_ season*_it_* + *b*_0*i*_ + *b*_1*i*_ *W_it_* + ε*_it_*

Non-linear model: *C_it_* = *β*_0_ + *f*(*W_it_*) + *β*_2_ season*_it_* + *b*_0*i*_ + *b*_1*i*_ *W_it_* + ε*_it_*

Here,

*C* is the average cadence;$\beta_{0}$ is the population-level intercept;$\beta_{1}$ is the fixed (linear) effect of the week number;*W* is the week number;$\beta_{2}$ is the fixed effect of the season (with the appropriate coding for categorical levels);$b_{0i}∼N(0,\sigma_{b0}^{2})$ is the random intercept for athlete i;$b_{1i}∼N(0,\sigma_{b1}^{2})$ is the random slope of the week number for athlete i;$\varepsilon_{it}∼N(0,\sigma^{2})$ is the residual error;$f(W_{it})$ is an unknown smooth function for the week number (the penalised smooth estimated by s(*W*)).

This statistical analysis was conducted using the mixed GAM computation vehicle (mgcv) package in R. Generalised additive modelling was selected because it does not require a pre-specified curve shape. Instead, the model estimates the form of the trajectory directly from the data. This is particularly valuable when analysing sports performance measures, where the underlying trend may change shape unpredictably and one should not rely on the analyst making assumptions about the type of curve involved [15]. Non-linear trends in average cadence were modelled using a thin plate regression spline. Participant-level variability was represented using random-effect smooths for the intercept and slope. Smoothness was penalised and estimated using restricted maximum likelihood (REML). In the mgcv package, the default thin plate regression spline basis was used to model the average cadence trend over time, whereas a random-effect spline basis was used to represent the random intercepts and random slopes. Model validation was assessed by comparing the relative fit, explained variance and model complexity using the Akaike information criterion (AIC), Bayesian information criterion (BIC), and coefficient of determination (r^2^).

Group-level predicted changes in average movement cadence were plotted with 95% confidence intervals and shaded, and individual-level predicted changes were plotted in different colours in another plot in the same figure (Figure 2, Figure 3 and Figure 4). For the participants who competed in two seasons, the GAM considered both seasons and we plotted a mean predicted change over time. Additionally, the estimated degrees of freedom (eDF) were calculated to quantify the extent of non-linearity in each non-linear model. While the individual plots appear to show all individuals beginning at the same time in week one, this was caused by the modelling of the data, and, in reality, each individual may have had their own ‘first week’ that was based on their first race of the season. Additionally, as participant-specific smooths were estimated as penalised random effects, individual trajectories were pooled towards the population mean, especially where participants contributed a low number of data points (Figure 4b—one participant contributed 3 races). This can result in modelled individual curves that do not pass through the raw data [15].

To highlight periods of time where the movement cadence changed the most, the percentage change was calculated between each of the weeks. Finally, the fit for the model was calculated and the r^2^ reported to measure the amount of variation in average cadence that the model explained. Changes in movement cadence over time were visually assessed according to Wood [15], who explains that, when using GAM methods, the smooth functions are composed of many basis functions that have different coefficients, so calculating an overall coefficient of the smooth function is not useful. Rather, the shape and function of the smoothed curve is the sum of the basis functions and their coefficients and should instead be interpreted along with the confidence intervals [15].

Finally, to compare the performance of linear and non-linear models, the Akaike information criterion (AIC) [30] and Bayesian information criterion (BIC) [31] were calculated, and a likelihood ratio test using a chi-squared distribution was conducted. Regarding the AIC, a lower AIC among the compared models indicates a better model with fewer parameters and a higher likelihood for the model. The Bayesian information criterion (BIC) was calculated to provide additional context for the comparison. It more harshly penalises the number of parameters compared to the AIC, leading to a better measure of model parsimony [31]. However, it is similar in that a lower number indicates better model performance.

## 3. Results

The three participants who raced in both seasons competed in approximately 12 races each (median = 12; range = 9–17), and participants who raced in one season competed in around five races each (median = 5; range = 3–8). Figure 1 details the recruitment and attrition of participants throughout the research.

To compare the linear and non-linear models, the AIC, BIC, amount of variance explained by each model for each discipline, and results of the likelihood ratio test to check for differences between models are reported in Table 1.

The findings reported in Table 1 suggest that a non-linear approach is more appropriate to model swimming cadence; however, in cycling, both models performed approximately equally. Finally, in running, the linear model appeared to perform better and with less complexity; however, there were no significant differences between the two. Group and individual changes in performance over time for each discipline were subsequently modelled using each approach and are reported in Figure 2, Figure 3 and Figure 4.

**Figure 2 sensors-26-00096-f002:**
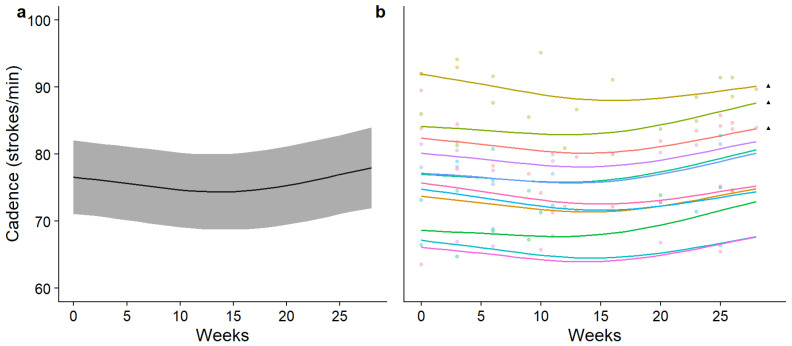
Modelled mean (±95% CI) time-normalised changes in swimming cadence over time (**a**) and individually modelled, time-normalised changes in swimming cadence over time (**b**). In part (**b**) of the figure, each coloured line represents a different participant and each ▲ denotes the participants who raced in two seasons.

**Figure 3 sensors-26-00096-f003:**
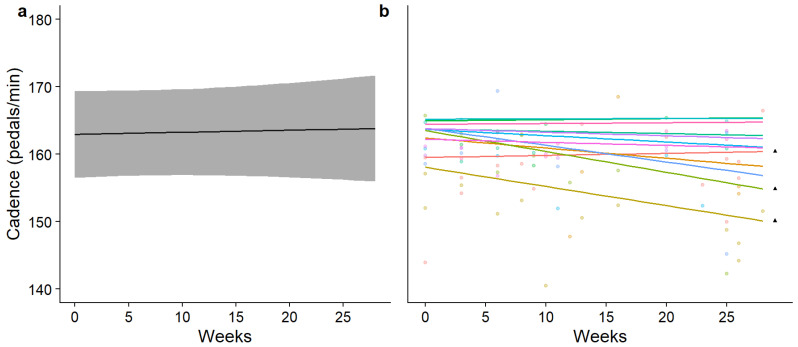
Modelled mean (±95% CI) time-normalised changes in cycling cadence over time (**a**) and individually modelled, time-normalised changes in cycling cadence over time (**b**). In part (**b**) of the figure, each coloured line represents a different participant and each ▲ denotes the participants who raced in two seasons.

**Figure 4 sensors-26-00096-f004:**
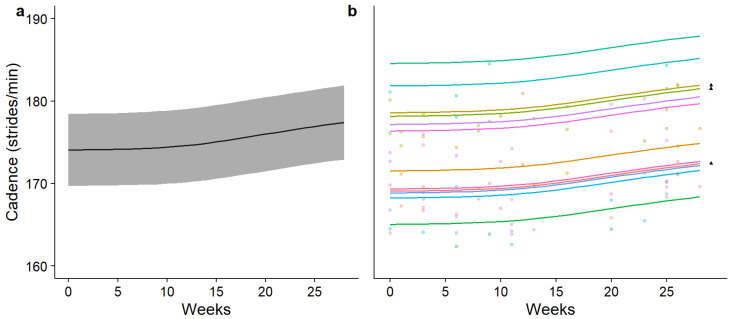
Modelled mean (±95% CI) time-normalised changes in running cadence over time (**a**) and individually modelled, time-normalised changes in running cadence over time (**b**). In part (**b**) of the figure, each coloured line represents a different participant and each ▲ denotes the participants who raced in two seasons.

In swimming, the results of the GAM predict that athletes experience a temporary decrease in swimming cadence during the season, which then begins to increase towards the end of the season (eDF = 4.3). In cycling, modelling predicts that there is very little common change in cycling cadence shared by participants and a linear change over time (eDF = 0.03); however, some participants appear to experience a substantial decrease over time. Finally, the model predicts a common increase in running cadence across the cohort as each season progresses, which varies mildly in trajectory as the season progresses (eDF = 2.07). The most accelerated change in predicted swimming cadence was measured between weeks 18 and 27, where the mean change in swimming was 0.41% per week. In cycling, the change in predicted cycling cadence was linear and uniform, changing by 0.02% per week for the entire season. Lastly, the most accelerated period of change in the predicted running cadence was also in the final phase of the season (weeks 14–27), where the running cadence increased by 0.10% per week on average.

## 4. Discussion

Non-linear regression showed increases in swimming and running cadence across the cohort, as well as individual differences in movement cadence between participants. When the individual movement cadence is observed (Figure 2b, Figure 3b and Figure 4b), a similar general function is observed across all participants, but some differences in the trajectory of change over time exist. Differences between linear and non-linear model performance, as well as the important applications of using a wearable IMUs to measure changes in triathlon movement cadence over time, will be discussed next.

A distinct and novel aspect of our reporting is the contrasting of the linear and non-linear modelling of longitudinal performance to measure human performance variables [32]. When observing the results of linear and non-linear modelling, there are significant differences between each modelling method in swimming cadence but not in cycling and running. In swimming, both the AIC and BIC measures show that the non-linear model provides a significantly better fit with less complexity than the linear model. However, there are no significant differences in model performance for cycling and running. The lack of significant differences between models in cycling and running could be explained by the near-linear trend even when non-linear modelling is performed in running, as well as the confounding effect of gearing in cycling. The superior performance of non-linear modelling compared with linear modelling in swimming highlights the importance of using theoretically correct analysis methods when analysing human performance data.

A key aim of triathlon training is to optimise the balance between the movement frequency (cadence) and distance per propulsive action (length). Thus, wearable IMUs provide a valid tool for monitoring cadence over time to guide triathletes towards this balance. The non-linear analysis of swimming cadence predicts that the cadence decreases mid-season before increasing again, with no race characteristic (e.g., race distance) explaining this pattern. A similar phenomenon was reported by Anderson et al. [33], who found that female swimmers had a decreased stroke rate and increased stroke length and velocity, while male swimmers exhibited an increased stroke rate and decreasing stroke length over time. We extrapolate the reporting by Anderson et al. [33] such that the triathletes reported here were learning to optimise their stroke rate and length over time to maximise their velocity, with this optimisation likely influenced by the pulling strength, possibly explaining the sex-based differences. Although confirming this phenomena would require intra-cycle velocity data, this interpretation aligns with findings that the swimming cadence remains stable early in adolescence and begins to change over time [34,35]. Taken together, this shows that, if the wrong modelling techniques are used, the insights gained may lead to training prescription errors or a periodisation adjustment for triathletes based on incorrectly modelled data, which may fail to drive performance improvements.

The non-linear modelling of cycling cadence indicated a lack of trend over time. Additionally, the substantially lower adjusted r^2^ and explained deviance indicate that time has a minimal impact on the observed variance in cycling cadence. Initially, this suggests that, as triathletes spend more time cycling, they do not learn to ride with higher cadences; however, it is more likely that changes in cycling cadence were concealed by confounding factors. For example, movement during cycling is possible without direct propulsion, which is not the case in swimming and running. In both swimming and running, the velocity (m/s) (the performance outcome) is equal to the movement frequency (Hz) multiplied by the length of each stride or stroke (in metres) [36,37]. In cycling, gearing systems allow cyclists to produce a large range of velocities at the same cadence, and the movement available while coasting (cycling without pedalling) makes evaluating the quality of cycling cadence more difficult. Variations in cycling cadence could also have been due to two complementary factors. Firstly, drafting during racing is legal, and, as drafting decreases the drag experienced by cyclists, this allows higher cadences to be achieved for the same work performed [38]. Second, while drafting, cyclists can cycle in a lower gear and increase their cycling cadence while maintaining similar velocities due to reduced drag, further confounding the inference of improvements in motor skill performance from changes in cycling cadence. To understand changes in cycling cadence more accurately, information about a cyclist’s gear selection is required.

Non-linear analysis predicts that the running cadence increases progressively across races within a season. Although differences in running cadence are likely due to age and experience, the shape of the function across participants over time was consistent, and the model explained a large proportion of the variance. The shape of the trend plateaus at the beginning and end of the season, with most improvements occurring mid-season. This pattern may reflect the racing schedule, with an even distribution of races between weeks 0 and 10, reduced racing opportunities between weeks 10 and 20 and some final races around week 20 and weeks 25–27. Gradual changes in running cadence, such as the one to two strides per minute observed across the season, are expected among highly trained, national-level athletes. However, these changes are small, remaining within the 95% confidence intervals of the model. This highlights the need to refine the model to accurately detect changes of this magnitude over the course of a season. Analysing data like these has practical value, as sports scientists can model improvements in a triathlete’s running cadence to support the pursuit of an optimal balance between stride frequency and length.

The practical application of modelling movement cadence data collected by a wearable IMU is that coaches can then apply targeted interventions based on this modelling, such as auditory feedback or attentional focus strategies [39,40]. This application highlights the integration of wearable IMU use in guiding the improvement of motor skill performance [41]. If any increases in movement cadence are due to learning, they are expected to be due to an improved ability to parameterise the GMP of swimming, cycling and running by scaling the speed at which the GMP is executed [17]. After races, participants update the parameters within the motor response schema through self-evaluation from intrinsic feedback and sensory information collected during the race, as well as feedback from the coach and information from the wearable IMU [17,42,43]. Improvements in motor skill adaptability are inferred if this results in improved performance from race to race over time.

This research is the first longitudinal analysis of swimming, cycling and running cadences in triathlons using statistical models that account for the non-linear nature of changes in motor skill performance. It employed a single commercially available IMU (every additional IMU used to capture data increases the complexity of synchronisation and analysis) and established data processing techniques [21], resulting in a proof of concept for modelling the changes in a specific motor skill over time. Modelling motor skill performance can allow practitioners and coaches to respond to changing performance over time by implementing more targeted training prescriptions. For example, a coach could measure the movement cadence of a triathlete across races to create a performance curve of changes over time [44]. Based on these performance curves, triathlon coaches could implement strategies like auditory feedback training to improve gait characteristics and promote a positive change in performance.

The length of time for which the triathletes were followed provided a 6–18-month ‘snapshot’ of their training careers (with ▲ denoting those who were followed for 18 months in Figure 2, Figure 3 and Figure 4). However, the average time spent competing in triathlons by the participants in this sample was 3.77 (±1.53) years, and training careers in total are likely to span more than a decade. This is particularly evident with the running movement cadence, where the somewhat linearly increasing trend over time may form only one part of a differently shaped trend that more accurately describes motor skill performance changes over the whole sporting career. Thus, caution is advised when extrapolating the changes in performance taken from this period and making assumptions about the entire training career. A much larger time scale might be required to gain a true picture of the overall performance changes of each participant.

One of the limitations of this study is that changes in the mean predicted cycling cadence (Figure 3a) were not observed across the two seasons, likely due to confounding factors masking potential changes. Cadence measurements are therefore better conducted in a controlled setting, such as on a cycling ergometer, where the conditions are standardised and cyclists can maintain consistent effort relative to their functional threshold power. Another limitation of this research is the inability to attribute a cause to the changes in movement cadence over time. Future research should therefore study longitudinal changes in movement cadence while also collecting information about confounding variables, such as race course characteristics (tidal conditions, competitor densities, elevation gain, number of corners); psychological readiness and motivation to compete; descriptions of internal (ratings of perceived exertion) and external training loads (training distances, durations and velocity zones); and measures of physical maturity, like the peak height velocity.

This investigation highlights the practical applications of this modelling approach by enabling the identification of triathletes who follow a typical improvement trajectory or who improve at atypical rates, allowing coaches to identify talent or provide targeted support for their triathletes. Second, this method of analysis offers a potential tool for evaluating the effectiveness of training strategies that aim to improve movement cadences. By modelling cadence trends over time, practitioners can assess whether specific interventions are translating into improved race performance in swimming and running, even in the presence of confounding variables such as varying race distances or differing environmental and race constraints. Third, the consistent monitoring and modelling of cadences longitudinally may assist in identifying early signs of fatigue, particularly where a decline in movement cadence occurs under otherwise stable training conditions. Such degradation in motor skill execution could signal accumulating fatigue or insufficient recovery and warrant modification to training loads or periodisation.

Future investigations should aim to build on this analysis by including multiple parameters of performance, such as swimming stroke and running stride lengths. Modelling multiple parameters of performance concurrently could provide greater insights into the optimal balance of movement frequency and cycle length. Additionally, applying neural networks to analyse movement cadence data may allow more flexible and accurate predictions of trends over time, rather than relying on the fit of a predefined curve. While the current investigation utilised an automatic peak counting algorithm to measure movement cadences, machine learning methods could be used to analyse the shapes of the accelerometer and gyroscope signals so as to detect potential abnormal patterns. It would be useful to practitioners to know if atypical or unusual signal patterns also reflect poor technique or abnormal movement.

The present investigation has shown that there are some significant differences found in triathlon movement cadence data when modelled with statistical methods that more closely reflect the nature of changes in performance in the real world. Broadly similar patterns of change over time are shown by individuals in swimming and running, with some exceptions in cycling; however, the overall cadences varied highly across individuals (Figure 2b, Figure 3b and Figure 4b). What remain to be understood are the contributions of maturation, learning and improvements in physiological fitness and strength to individual differences in movement cadence over a triathlon season. Further investigations should also validate the measurement of other important triathlon motor skills, such as coasting during cycling [20], and posture in each discipline [3]. Finally, to address the difficulties in evaluating cycling cadence, the integration of this wearable sensor with an electronic gearing system would enable the combination of gearing information with cadence, providing more informative performance metrics.

## Figures and Tables

**Figure 1 sensors-26-00096-f001:**
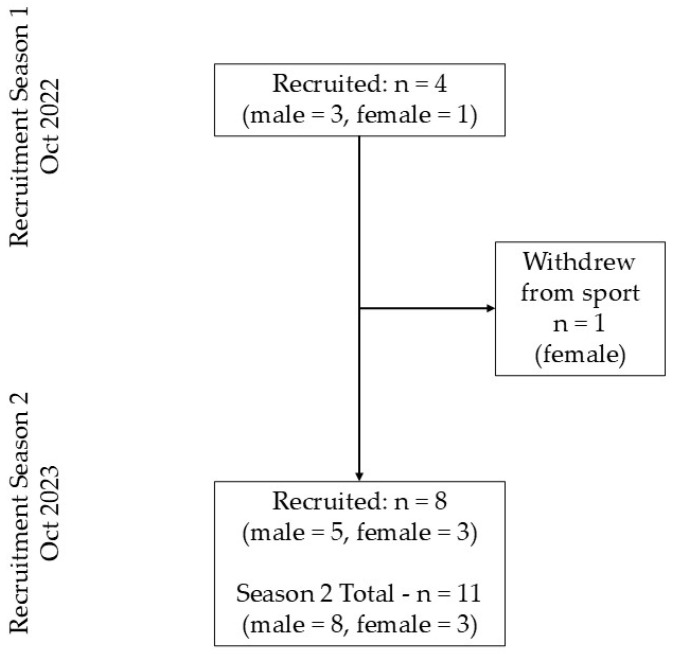
Participant recruitment/attrition flow chart.

**Table 1 sensors-26-00096-t001:** Comparison of model performance for each triathlon discipline.

Discipline	Model Type	AIC	BIC	r^2^	*D*	*p*
Swimming	Linear	426.39	474.85	0.88	-	-
	Non-Linear	413.61	470.15	0.90	106.71	<0.01
Cycling	Linear	537.46	578.11	0.31	-	-
	Non-Linear	537.51	578.22	0.31	0.05	0.08
Running	Linear	392.22	428.59	0.86	-	-
	Non-Linear	391.14	431.50	0.87	16.71	0.17

## Data Availability

The raw data supporting the conclusions of this article will be made available by the authors on request.

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
