# Peer review of "A Longitudinal Analysis of a Motor Skill Parameter in Junior Triathletes from a Wearable Sensor"

_sensors, 2025, doi:10.3390/s26010096_

Round 1

Reviewer 1 Report

Comments and Suggestions for Authors

This study proposes a customized automatic peak detection algorithm to analyze the movement cadence data of triathletes. However, several aspects require further clarification, which are outlined in the following comments:

  1. In the Introduction, the algorithm used for peak detection is not adequately introduced, and the summary of previous research on movement cadence analysis algorithms is insufficient. The overview of existing nonlinear analytical methods is also limited. The authors are encouraged to clarify which nonlinear analysis method was employed, why this specific method was chosen, and what limitations exist in previous studies on movement cadence algorithms.
  2. More recent reports on advanced wearable sensors for monitoring human health and exercise (e.g., doi.org/10.59717/j.xinn-mater.2025.100143) are useful and could be included for enriching the research background.
  3. There appear to be highlighted yellow sections in the manuscript, which may indicate an editing oversight. In addition, within Section 2, the proposed automatic peak detection algorithm is not described in sufficient detail. It is recommended that the authors include a schematic diagram to better illustrate the algorithmic workflow.
  4.  In Figures 2–4, the authors only briefly present the cadence analysis results for the triathlon disciplines. However, the figures lack units and legends. Furthermore, the statistical analysis is relatively limited and should be expanded to enhance the robustness of the results.
  5. The Discussion section should be more concise. Moreover, since both linear and nonlinear Generalized Additive Models (GAMs) were used for analysis, the manuscript should include the mathematical formulations or model equations for these two approaches to improve methodological transparency.

Reviewer 2 Report

Comments and Suggestions for Authors

This article presents the results of a longitudinal study of junior triathletes (n = 12) monitored over two seasons. The aim was to assess the evolution of movement cadence (a motor parameter) using wearable inertial sensors (IMU – Catapult Optimeye S5).
The study aimed to compare the performance of linear and nonlinear (general additive model, GAM) models in describing the temporal behaviour of cadence in three sports: swimming, cycling and running.
The authors report that nonlinear models provided a better fit for swimming and running, whereas for cycling there was no significant difference. The results suggest that the motor development of young athletes occurs in a non-linear and adaptive manner, thereby reinforcing the usefulness of wearable sensors in motor analysis.

Suggested decision: Major revision

The article has high scientific potential, particularly given its analytical approach and the relevance of the topic. However, substantial revisions to the methodology and presentation of the results are required to achieve the level of rigour necessary for publication.

Suggestions:

  1. It is recommended that more recent references to the longitudinal use of IMUs in young athletes are included.
  2. Rewrite the hypothesis section in the introduction to make it more explicit.
  3. Technical information about the sensors (calibration, accuracy and validation) should be included.
  4. The type of smoothing function, adjustment parameters, spline selection method and model validation should be described more clearly.
  5. Add confidence intervals and effect sizes to the results.
  6.  Restructure the discussion to reduce redundancies and expand on the practical applications.
  7.  Present future perspectives, such as analysis with neural networks or integration of multiple parameters (e.g. cadence, speed and heart rate).
  8. The figures are well organized, but could highlight inflection points and regions of greater variation to make them easier to read.

Round 2

Reviewer 1 Report

Comments and Suggestions for Authors

The manuscript has been improved according to the comments.

Reviewer 2 Report

Comments and Suggestions for Authors

Once these major revisions have been made, I recommend accepting the manuscript.
